# A RISK-AVERSE EQUILIBRIUM FOR MULTI-AGENT SYSTEMS

## ABSTRACT

In multi-agent systems, intelligent agents are tasked with making decisions that lead to optimal outcomes when actions of the other agents are as expected, whilst also being prepared for their unexpected behaviour. In this work, we introduce a novel risk-averse solution concept that allows the learner to accommodate low probability actions by finding the strategy with minimum variance, given any level of expected utility. We first prove the existence of such a risk-averse equilibrium, and propose one *fictitious-play* type learning algorithm for smaller games that enjoys provable convergence guarantees in games classes including zero-sum and potential. Furthermore, we propose an approximation method for larger games based on iterative population-based training that generates a population of risk-averse agents. Empirically, our equilibrium is shown to be able to reduce the utility variance, specifically in the sense that other agents' low probability behaviour is better accounted for by our equilibrium in comparison to playing other solutions. Importantly, we show that our population of agents that approximate a risk-averse equilibrium is particularly effective against unseen opposing populations, especially in the case of guaranteeing a minimum level of performance, which is critical to safety-aware multi-agent systems.

## 1 INTRODUCTION

Game Theory (GT) has become an important analytical tool in solving Machine Learning (ML) problems; the idea of "gamification" has become popular in recent years (Wellman, 2006; Lanctot et al., 2017) particularly in multi-agent systems research. The importance of risk-aversion in the single-agent decision making literature (Zhang et al., 2020; Mihatsch & Neuneier, 2002; Chow et al., 2017) is obvious, whilst there still exist many open questions in the current game theory research domain. This paper aims to add to the current research in the multi-agent strategic decision-making literature based on the notion of risk-aversion through the lens of a new equilibrium concept.

One reason that risk-aversion is important is that multi-agent interaction is rife with strategic uncertainty; this is because performance doesn't solely depend on ones own action. It is rarely the case that one will have certainty over the execution and the strategy of the opponent in situations ranging from board games to economic negotiations (Calford, 2020). This presents a dilemma for autonomous decision-makers in human-AI interaction as one can no longer rely on perfect execution or complete strategy knowledge. Therefore, an important issue is what happens when actors take dangerous low probability actions such that could be considered as mistakes. These issues in play can arise in an array of circumstances, from misunderstandings of reward structures to execution fatigue, leading to the execution of an unexpected pure strategy. Hedging against unexpected play is important for the agents as otherwise it can lead to large costs. As demonstrated in Fig. (1), a mistake in the execution of the pure-strategy Nash equilibrium (NE) could lead to both cars overtaking and crashing into each other, a negative yet critical outcome in multi-agent system.

Traditional equilibrium solutions in GT (e.g. NE, Trembling Hand Perfect Equilibrium (THPE) (Bielefeld, 1988)) lack the ability to handle this style of risk as either: 1) they assume strategies are executed perfectly, and/or, 2) large costs may be undervalued if there is a low probability attached to them. We address these by introducing a new framework for studying risk in multi-agent systems through mean-variance analysis. In our framework, strategies are evaluated both in terms of expected utility against the opponent, but also the potential utility variance if the opponent played

|  | Stay in Lane | Overtake |
|---|---|---|
| Stay in Lane | 5, 5 | 0, 20 |
| Overtake | 20, 0 | -50, -50 |

Pure Strategy Nash Equilibrium

Risk Averse Equilibrium

Figure 1: Cars are rewarded for reaching their destination. They are behind slow tractors but can stay in their lanes and arrive safely, but slowly. They can overtake to arrive quickly, but if the other also overtakes they will crash, leading to large negative payoffs. The *Overtake* strategy is high-risk, high-reward and susceptible to errors, and is selected under a Nash equilibrium. The *Stay in Lane* strategy is low-risk, low-reward with low variance and selected by our mean-variance RAE approach.

low probability pure strategies. For example, the driving example in Fig. (1) describes a simple scenario where, due to the critical nature of wanting to avoid crashing, the benefits of overtaking may be entirely redundant with the possibility of low probability play leading to crashes. We summarise the contributions of our paper here:

1. We introduce a novel risk-averse equilibrium (RAE) based on mean-variance components of the available strategies. Our framework generalises the single-agent mean-variance decision framework to multi-agent settings.

2. We show that the RAE always exists in finite games, and that it is solvable in the class of games with the fictitious-play property. This, as we later show, unlocks a powerful array of computational methods for solving games.

3. We demonstrate that: 1) RAE is able to locate a minimum variance solution for any given level of utility 2) A by-product of RAE is that it can be used as a Nash equilibrium selection tool in the presence of a "risk-dominant" equilibrium 3) RAE is able to find a low risk strategy in a safety-sensitive autonomous driving environment.

## 2 RELATED WORK

There exists three relevant bodies of work, those works that empirically study the presence of risk-aversion in humans, those that aim to develop new equilibrium frameworks and those that study how risk-averse agents alters classical game-theoretic results.

On the empirical side, the first paper to show that humans prefer to bet on known probability devices, rather than on other human choices, suggesting strategic uncertainty avoidance (Camerer & Karjalainen, 1994). Bohnet & Zeckhauser (2004) similarly found that subjects are more trusting in an objective randomisation device rather than other humans. Eichberger et al. (2008) found that more trust is placed in game theorists than "grannies" as the latter is a source of strategic ambiguity. Similar practices are noted in the game setting which more closely model multi-agent interactions, especially in the form of ambiguity aversion, such as those games outside of 3-color Ellsberg Urn tasks (Kelsey & Le Roux, 2015), public goods and weakest link games (Kelsey & Le Roux, 2017), or in the presence of strategic complements and strategic substitutes (Kelsey & le Roux, 2018). For an extensive survey of the experimental evidence, we refer readers to (Harrison & Rutström, 2008).

The equilibrium literature can be divided into three distinct sections. Harsanyi et al. (1988) introduced risk-dominant Nash equilibria (NE) (Nash, 1951), which suggests that increasing levels of strategic ambiguity will lead to the equilibrium with the lowest deviation losses. Risk dominance is limited as it is restricted to the set of NE strategies, and therefore may be risk-dominant in comparison to other NE but not particularly risk-averse at all. Bielefeld (1988) set out the THPE which deals with strategic risk by accounting for off-equilibrium play. However, this is sensitive to strictly dominated strategies and, because all trembles happen with marginal probability large downside risk can be masked, (In Fig.1 an error probability of 0.01 would only impact the utility function by 0.5, whereas we later show that our variance solution values the error at $47.6\gamma$, where $\gamma > 0$ and non-marginal), which is problematic for safety-sensitive systems (e.g., autonomous driving). McKelvey & Palfrey (1995) utilises the Quantal Response Equilibrium (QRE) to introduce errors into strategy selection, but with lower percentages on big mistakes which also discounts the impact of large downside risk. We argue that QRE undervalues big costs which are particularly damaging in real-world settings, whereas

our mean-variance approach hedges away from high cost risk even in the presence of marginal probabilities. Yekkehkhany et al. (2020) utilises a similar mean-variance equilibrium concept based on risk derived from one-shot play in the probabilistic setting, rather than the expectation setting of this work. This does not apply more generally to the model-free machine learning setting where utility probabilities are not known and is therefore more difficult to apply practically.

In terms of risk-aversion outside of equilibrium concepts, competitive network games Wardrop (1952) and the non expected utility maximising setting Fiat & Papadimitriou (2010) have been studied the most. Risk aversion in the network setting is based on a generalisation of the classical selfish routing model Beckmann et al. (1956) to incorporate uncertain delays. Nikolova & Stier-Moses (2014) consider a mean-variance framework for Wardrop equilibria in this setting, whilst Lianeas et al. (2019) extend this research to looking at how risk aversion degrades the performance of a routing system. Whilst the mean-variance approach is the same underlying notion as our work, we instead propose a solution for general games rather than routing games. In general games, Fiat & Papadimitriou (2010) remove the assumption of expectation maximisation and show that under risk averse utility functions there may exist no Nash equilibria. Further works have generally focused on Price of Anarchy Piliouras et al. (2016); Kesselheim & Kodric (2018), which study how removing the assumptions of risk neutrality in general games impacts the difference between the achieved worst equilibrium and the maximum possible welfare of the system. Our work follows a similar strand in looking at general games, but focuses on defining a new equilibrium concept rather than establishing how risk-averse agents change the convergence properties to classical equilibrium concepts. In addition, we frame our work such that it is more scalable for usage alongside RL techniques.

Our framework fits broadly into the areas of risk-sensitive RL ((Chow & Ghavamzadeh, 2014; Keramati et al., 2020; Zhang et al., 2021)) and risk measures, such as mean-variance, value at risk (VaR) and conditional value at risk (CVaR), in RL ((García & Fernández, 2015; Tang et al., 2019; Hiraoka et al., 2019; Ma et al., 2020)). The focus of risk-sensitive RL has remained predominately in single-agent settings where risk is due to the environment, rather than from other inhabitants of the environment. Multi-agent solutions include: RMIX (Qiu et al., 2021) which optimises decentralised CVaR policies in cooperative risk-sensitive settings, RAM-Q and RA3-Q (Gao et al., 2021) which tackles algorithmic trading by utilising an adversarial approach to promote variance reduction, or risk-sensitive DAPG (Eriksson et al., 2022) which approaches risk in Bayesian games in terms of the CVaR induced by the possible combinations of types in the game. However, as we are specifically concerned with game-theoretic equilibrium concepts we will not directly compare to these methods.

Historically, the key challenge of computational GT is how to solve for a NE. For example, in two-player zero-sum games, it is theoretically possible to solve for an NE directly via linear programming (LP) (Morgenstern & Von Neumann, 1953). Another approach to finding an equilibrium is the iterative method Fictitous Play (FP) (Brown, 1951), where players make best-responses to the time-average action of the opponent. However, in practice both the above approaches can be strictly intractable. Limitations due to action space size led to a general wave of methods that focus on starting with a "restricted" action space and iteratively expanding said space in order to approximate an equilibrium with the best possible strategies. Notably, Double Oracle (DO) (McMahan et al., 2003; Dinh et al., 2021; McAleer et al., 2021) and Policy-Space Response Oracles (PSRO) (Lanctot et al., 2017; McAleer et al., 2020; Perez-Nieves et al., 2021; Feng et al., 2021) methods are the two major frameworks in this area. In this work we face a similar challenge in terms of the difficulty of solving for our own equilibrium. In this paper we demonstrate how FP and PSRO can be applied as a solver for our new equilibrium concept. In doing so, we provide a concrete methodology for obtaining solutions in our setting. However, we must adapt them as they are generally designed for risk-neutral equilibria which is not the case for this work.

## 3 PRELIMINARIES & NOTATIONS

In this section, we introduce the preliminaries and notation required to understand our formulation. A normal-form game (NFG) is the standard representation of strategic interaction in GT. A finite $n$-person NFG is a tuple $(N, A, u)$, where $N$ is a finite set of $n$ players, $A = A^1 \times, ..., \times A^n$ is the joint action profile, with $A^i$ being the strategies available to player $i$, and $u = (u^1, ..., u^n)$ where $u^i : A \to \mathbb{R}$ is the real-valued expected utility function for each player. A player plays a mixed-strategy, $\boldsymbol{\sigma}^i \in \Delta_{A^i}$, which is a probability distribution over their possible actions. In Sec. 6 we

replace our atomic pure strategies with neural network based strategies and therefore re-define our notation to keep clarity between the two game schemes.

The central equilibrium concept in game-theory is the Nash equilibrium (NE), which is a strategy profile where no players have an incentive to deviate. Let $a_{-i} \in A_{-i}$ be the pure strategy sets for all players other than $i$. Let $u_i(a_i, a_{-i})$ be the expected utility function for player $i$ versus all players other than $i$, the strategy profile $a^* = (a_i^*, a_{-i}^*)$ is a NE if,

$$u_i(a_i^*, a_{-i}^*) \geq u_i(a_i, a_{-i}^*) \quad \forall a_i \in a_i \tag{1}$$

## 4 MEAN-VARIANCE EQUILIBRIUM

In the following section, we introduce our mean-variance based total utility function and then show how it can be used as an equilibrium concept. Our proposed variance method aims to deal with the main downside of QRE and THPE, that they both undervalue large downsides. For example, QRE is designed that action probabilities are proportional to expected utility and THPE assigns 'error' probability to all zero-probability actions. In both of these cases, variance from the average expected utility will provide a more pronounced effect of risk than using the raw values.

### 4.1 UTILITY FUNCTION

Here we propose a total utility function that measures both the expected utility, but also the potential variance of utility dependent on the opponent's strategy.

For simplicity we provide definitions based on playing a symmetric game, such that two players share an action set $\mathcal{A}$, a utility function $u$. We extend this to the non-symmetric case in Appendix H. Define the expected utility of action $a_i \in \mathcal{A}$ against action $a_j \in \mathcal{A}$ as $u(a_i, a_j)$ and the full expected utility table as $\mathbf{M}$, where the entry $\mathbf{M}_{i,j}$ refers to $u(a_i, a_j)$ and $\mathbf{M}_i$ refers to $u(a_i, a_j) \,\forall j$, i.e. the vector of expected utilities that action $a_i$ receives against all other actions. We now define the expected utility of the mixed-strategy for player 1 $\boldsymbol{\sigma}$ versus the mixed strategy for player 2 $\boldsymbol{\varsigma}$ as

$$u(\boldsymbol{\sigma}, \boldsymbol{\varsigma}, \mathbf{M}) = \sum_{a_i \in A} \sum_{a_j \in A} \sigma(a_i)\varsigma(a_j)u(a_i, a_j) = \boldsymbol{\sigma}^T \cdot \mathbf{M} \cdot \boldsymbol{\varsigma}. \tag{2}$$

The weighted co-variance matrix for $\mathbf{M}$ is a $|A| \times |A|$ matrix $\boldsymbol{\Sigma}_{\mathbf{M}, \boldsymbol{\varsigma}} = [c_{ij}]$ with entries

$$c_{jk} = \sum_{a_i \in A} \varsigma(a_i)\big(u(a_i, a_j) - \bar{\mathbf{M}}_j\big)\big(u(a_i, a_k) - \bar{\mathbf{M}}_k\big), \tag{3}$$

where $\bar{\mathbf{M}}_i = \frac{1}{|A|} \sum_{k=1}^{|A|} \varsigma_k u(a_i, a_k)$ is the weighted average expected utility for action $i$. This is a standard co-variance matrix where the values for each action are weighted by the likelihood of them being selected by the opponent. A uniform weighting could be used, however we believe that in terms of utility variability avoidance it is more intuitive to hedge against the variance caused by high likelihood actions. However, as will be discussed later, all actions will still receive positive probability under our framework and therefore will always provide some weight in the variance calculation, leading to low likelihood high-variance actions still having a large impact on the final result. This accounts for the idea that mistakes may happen such that all actions can be played with at least a low probability. This allows us to define the mixed-strategy $\boldsymbol{\sigma}$ variance utility as follows:

$$\mathrm{Var}(\boldsymbol{\sigma}, \boldsymbol{\varsigma}, \mathbf{M}) = \sum_{k=1}^{|A|} \sum_{n=1}^{|A|} \sigma(a_k)\sigma(a_n)c_{kn} = \sum_{i=1}^{|A|} \sigma(a_i)^2 c_{ii} + \sum_{k=1}^{|A|} \sum_{n=k+1}^{|A|} \sigma(a_k)\sigma(a_n)c_{kn}$$
$$= \boldsymbol{\sigma}^T \cdot \boldsymbol{\Sigma}_{\mathbf{M}, \boldsymbol{\varsigma}} \cdot \boldsymbol{\sigma}. \tag{4}$$

The final total utility function $r$ which considers expected and variance utility for strategy $\boldsymbol{\sigma}$ is,

$$r(\boldsymbol{\sigma}, \boldsymbol{\varsigma}, \mathbf{M}) = u(\boldsymbol{\sigma}, \boldsymbol{\varsigma}, \mathbf{M}) - \gamma \mathrm{Var}(\boldsymbol{\sigma}, \boldsymbol{\varsigma}, \mathbf{M}), \tag{5}$$

where $\gamma \in \mathbb{R}$ is the risk-aversion parameter. Applying Eq. (5) to Fig. (1) we show how we arrive at a strategy profile that has our desired properties. Consider two joint strategy profiles,

$\boldsymbol{S}_1 = ((1-\epsilon, 0+\epsilon), (1-\epsilon, 0+\epsilon))$ and a Nash equilibrium $\boldsymbol{S}_2 = ((0+\epsilon, 1-\epsilon), (1-\epsilon, 0+\epsilon))$ where $(1-\epsilon, 0+\epsilon)$ represents playing *Stay in Lane* with probability $(1-\epsilon)$. $\epsilon$ is arbitrarily small and used to ensure fully mixed strategies, for the example we use $\epsilon = 0.01$. Profile $\boldsymbol{S}_1$ receives $u(\boldsymbol{S}_1) = 5$ and the Nash profile receives $u(\boldsymbol{S}_2) = 20$. However, $\mathrm{Var}(\boldsymbol{S}_1) = 0.32$ and $\mathrm{Var}(\boldsymbol{S}_2) = 47.6$, i.e. the Nash strategy has huge variance for Player 1. Therefore, $r(\boldsymbol{S}_1) = 5 - 0.32\gamma$ and $r(\boldsymbol{S}_2) = 20 - 47.6\gamma$ and we have for any risk-aversion parameter $\gamma > 0.32$ it is optimal to play $\boldsymbol{S}_1$.

## 4.2 EQUILIBRIUM CONCEPT

We now define our new equilibrium concept based on the total utility function (5). First start by defining the best-response map:

$$\boldsymbol{\sigma}^*(\varsigma) \in \arg\max_{\boldsymbol{\sigma}} r(\boldsymbol{\sigma}, \varsigma, \mathbf{M})$$
$$\text{s.t. } \sigma(a) \geq 0 \,, \forall a \in A \tag{6}$$
$$\boldsymbol{\sigma}^T \mathbf{1} = 1,$$

where due to the quadratic term $\boldsymbol{\sigma}^T \cdot \Sigma_{\mathbf{M},\varsigma} \cdot \boldsymbol{\sigma}$ and the constraints, we have a Quadratic Programme (QP). The above programme finds $\boldsymbol{\sigma}$ such that the total utility is maximised, whilst ensuring no strategies are assigned negative action probability, and that the action probabilities sum to one. We now propose the following:

**PROPOSITION 1.** *For any given expected utility $\mu_b$, there exists $\gamma$ such that the solution to (6) is the minimum variance solution.*

We defer the proof to Appendix (A). This proposition implies that when using 6, given any expected utility value $\mu_b$, there exists $\gamma$ that achieves $\mu_b$ with the minimum possible variance. Therefore, $\gamma$ can be tuned to provide a desired expected utility whilst giving the user the minimum viable variance solution. Therefore, based on Eq. (6), we define the equilibrium for the strategy profile $\boldsymbol{\sigma}$,

**DEFINITION 2** (Risk-Averse Equilibrium (RAE)). *A strategy profile $\{\boldsymbol{\sigma}, \varsigma\}$ is a risk-averse equilibrium if both $\boldsymbol{\sigma}$ and $\varsigma$ are risk-averse best responses to each other.*

Finally, a property of most game-theoretic equilibria is that a solution exists, at least in the finite game setting. For our equilibrium, we note the following result in mixed-strategies:

**THEOREM 3.** *For any finite N-player game where each player $i$ has a finite $k$ number of pure strategies, $A^i = \{a_1^i, ..., a_k^i\}$, an RAE exists.*

We defer the proof of the result to Appendix (A). Importantly, Theorem 3 establishes the existence of solutions providing practical relevance for our equilibrium concept.

## 5 EQUILIBRIUM LEARNING VIA STOCHASTIC FICTITIOUS PLAY

We start by showing that our total utility function can be used as a form of *stochastic fictitious play (SFP)* (Fudenberg & Kreps, 1993) for finding an RAE in small NFGs. SFP has convergence guarantees in a selection of games, most notably potential games (Monderer & Shapley, 1996a;b) and finite two-player zero-sum games (Robinson, 1951). Furthermore, SFP is robust to games outside of the above game classes (Ganzfried, 2020), and we extend these observations in Appendix (B).

SFP describes a learning process where each player chooses a best response to their opponents' time-average strategies. In *SFP*, a group of $n \geq 2$ players repeatedly play a $n-$player NFG. The state variable is $Z_t \in \Delta_S$, whose components $Z_t^i$ describe the time averages of each player's behaviour,

$$Z_t^i = \frac{1}{t} \sum_{u=1}^{t} \boldsymbol{\sigma}_t^i$$

where $\boldsymbol{\sigma}_t^i \in \Delta_{A^i}$ represents the observed strategy of player $i$ at time-step $t$. A *SFP* process is one where each player best responds to the time-average strategy of their opponent, $Z_t^{-i}$ such that,

$$\boldsymbol{\sigma}_{t+1}^i \in \arg\max_{\boldsymbol{\sigma}} u^i(\boldsymbol{\sigma}, Z_t^{-i}, \mathbf{M}) - \lambda v^i(\boldsymbol{\sigma}) \tag{7}$$

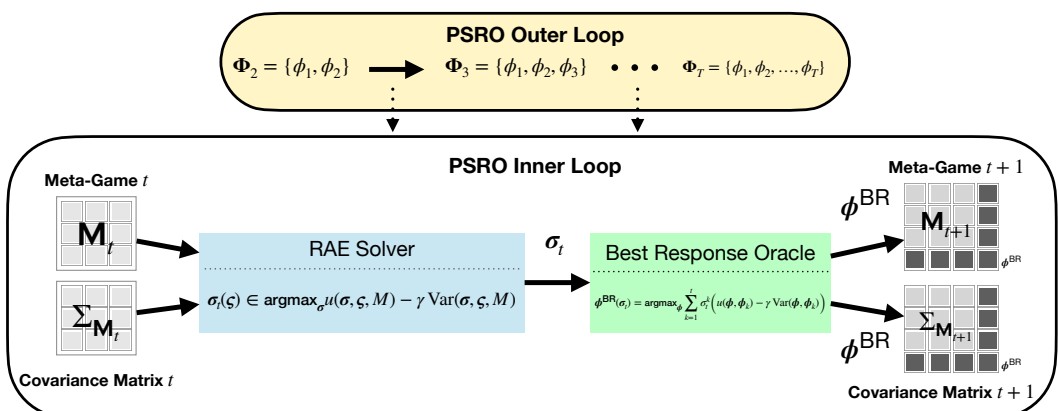

Figure 2: Iterative agent generation process. Note that $u(\cdot)$ and $\text{Var}(\cdot)$ are overloaded to represent utility/variance between distributions over a population or utility/variance between two policies.

$v^i(\boldsymbol{\sigma}) : \Delta_A \to \mathbb{R}$ is a strictly convex function, and the gradient of $v^i(\boldsymbol{\sigma})$ becomes arbitrarily large near the boundary of the strategy simplex, i.e. $\lim_{\boldsymbol{\sigma} \to \partial(A^i)} |v^i(\boldsymbol{\sigma})| = \infty$. We propose the following with regards to our total utility function (Proofs are deferred to Appendix (A))

**THEOREM 4.** *Given the total utility function Eq. (5) there exist RAE convergence guarantees in the category of games that are solved by SFP.*

SFP does not necessarily converge in all game classes (but is robust empirically). Therefore, we show that if the SFP process does converge to a strategy then that strategy is guaranteed to be an RAE,

**PROPOSITION 5.** *Suppose the SFP sequence $\{Z_t\}$ converges to $\boldsymbol{\sigma}$ in observed strategies [1], then $\boldsymbol{\sigma}$ is a risk-averse equilibrium.*

Note that for SFP we require a stronger notion of convergence in observed strategies $\boldsymbol{\sigma}_t^i$ rather than in beliefs $Z_t^i$, but the usage of a converged final $\boldsymbol{\sigma}_t^i$ guarantees a risk-averse equilibrium.

## 6 EQUILIBRIUM LEARNING VIA ITERATIVE AGENT GENERATION

For games that can't be tractably displayed in the normal-form, we use iterative solution frameworks, which make use of reinforcement learning (RL) policies as proxies for actions. This approach aims to approximate equilibria in large games by finding a small representative collection of risk-averse policies which can instead be selected over by RAE. We provide a visualisation of the following iterative agent generation process in Fig. (2), and provide an algorithm in Appendix D.

Consider two-player stochastic games $\boldsymbol{G}$ defined by the tuple $\{\mathcal{S}, \mathcal{A}, \mathcal{P}, \mathcal{R}\}$, where $\mathcal{S}$ is the set of states, $\mathcal{A} = \mathcal{A}^1 \times \mathcal{A}^2$ is the joint action space, $\mathcal{P} : \mathcal{S} \times \mathcal{A} \times \mathcal{S} \to [0, 1]$ is the state-transition function and $R^i : S \times \mathcal{A} \to \mathbb{R}$ is the reward function for player $i$. An *agent* is a policy $\phi$, where a policy is a mapping $\phi : \mathcal{S} \times \mathcal{A} \to [0, 1]$ which can be described in both a tabular form or as a neural network. The expected utility between two *agents* is defined to be $M(\phi_i, \phi_j)$ (i.e., in the same manner defined for NFGs in Sec. 4.1), and represents the expected utility to agent $\phi_i$ against opponent $\phi_j$.

Our iterative framework does $T \in \mathbb{N}^+$ iterative updates on a meta-game $\mathbf{M}$ (an NFG made up of RL agents as actions) following the framework of PSRO (Lanctot et al., 2017). At every iteration $t \leq T$, a *player* is defined by a population of fixed *agents* $\boldsymbol{\Phi}_t = \boldsymbol{\Phi}_0 \cup \{\phi_1, \phi_2, ..., \phi_t\}$, where $\boldsymbol{\Phi}_0$ is the initial random agent. For notation convenience, we consider the single-population case where players share the same $\boldsymbol{\Phi}_t$. As such, the population will generate a *meta-game* $\mathbf{M}_t$, an expected utility matrix between all the *agents* in the population, with individual entries $M(\phi_i, \phi_j) \, \forall \phi_i, \phi_j \in \boldsymbol{\Phi}_t$.

To make use of a population $\boldsymbol{\Phi}_t$ we require a way to select which agents $\phi_t \in \boldsymbol{\Phi}_t$ will be utilised for training. This function $f$ is a mapping $f : \mathbf{M}_t \to [0, 1]^t$ which takes as input a meta-game $\mathbf{M}_t$ and outputs a *meta-distribution* $\boldsymbol{\sigma}_t = f(\mathbf{M}_t)$. The output $\boldsymbol{\sigma}_t$ is a probability assignment to each *agent* in the population $\boldsymbol{\Phi}_t$ and, as we are in the *single*-population setting (i.e., symmetric play), we do not distinguish between populations. This is the equivalent of a mixed-strategy in a NFG, except now the

---

[1] Convergence in the time-average $Z_t$ does not imply convergence in the actual strategy taken at each $t$, but may for example imply cyclic actual behaviour that results in average behaviour converging.

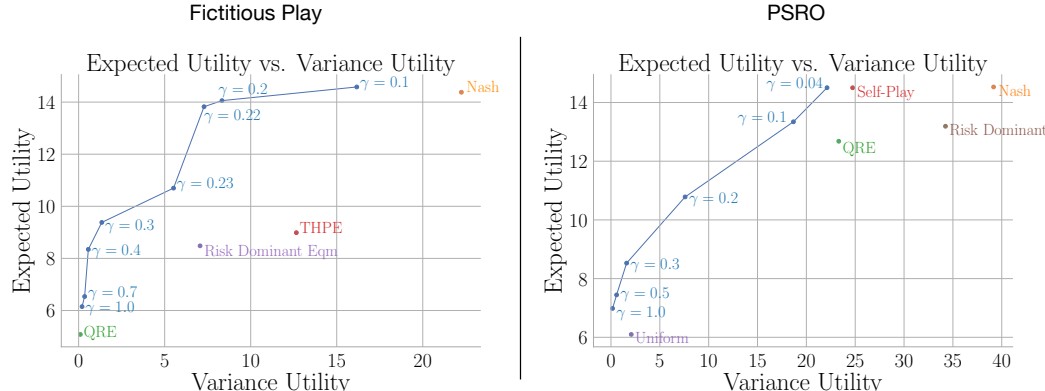

Figure 3: a) SFP on NFGs with 100 actions, b) PSRO on NFGs with 500 actions. Both compare final expected utility vs. variance utility results. We plot RAE values for multiple $\gamma$ to form an 'efficient frontier' and show that, whilst baselines achieve similar expected utilities they are always finding solutions that are too high in variance utility. In Fig. a) we exclude the Payoff Dominant result as it provided huge final variance utility, whilst in Fig. b) we exclude the THPE result for the same reason.

actions are RL policies. We apply our risk-averse equilibrium concept (Def. 2) as the meta-solver. As $\phi$ are RL policies then the policies are sampled by their respective probability in $\boldsymbol{\sigma}_t$.

At each epoch the population $\boldsymbol{\Phi}_t$ is augmented with a new agent that is a *best-response* to the meta-distribution $\boldsymbol{\sigma}_t$. Generally, this will be purely in terms of the expected reward, and can be found with any optimisation process such as Reinforcement Learning. In our setting there are two properties that we are concerned with when adding a new *agent* to the population. Notably, how it impacts the expected return but also how it impacts the variance utility of the population $\boldsymbol{\sigma}_{t+1} \cdot \boldsymbol{\Sigma}_{\mathbf{M}_{t+1}, \boldsymbol{\sigma}_{t+1}} \cdot \boldsymbol{\sigma}_{t+1}$.

To do this, we follow the PPO approach of (Zhang et al., 2020) that optimises both performance and per-step RL-reward variance. It is shown by (Bisi et al., 2019) that the variance of the per-step RL-reward bounds the variance of the total-RL reward from above. Notably, the variance utility of a population is measured in terms of the variance of the total RL-reward, and therefore shrinking the variance of the per-step RL-reward will also shrink the variance of the total RL-reward. To achieve this, an augmented MDP is used where the MDP-reward, $g_t^i$ is replaced as follows:

$$\hat{g}_t^i = g_t^i - \lambda(g_t^i)^2 + (2\lambda g_t^i y_i)$$

where $y_i = \frac{1}{T} \sum_{t=1}^{T} g_t^i$ is the average of the RL-rewards during the data collection phase. Notably, as this variance is also with respect to the sampling probability defined by $\boldsymbol{\sigma}_t$ this optimises the correct co-variance matrix which is similarly weighted by $\boldsymbol{\sigma}_t$.

## 7 EXPERIMENTS

We validate the effectiveness of RAE on three environments that all display some risk component.

1. Randomly generated coordination games where some actions provide a high expected utility if the other player selects the same action, but have large costs if not. There also exist actions that have lower coordinated expected utility but lower costs. We conduct experiments testing SFP on games with 100 actions, and utilising our iterative approach on games with 500 actions. Vanilla policy gradient RL agents are used for the iterative approach.

2. A generalised grid-world stag-hunt (Peysakhovich & Lerer, 2017) game that has a payoff-dominant and risk-dominant equilibrium. In this game it is not possible tractably to list out all actions and therefore our iterative approach is applied. PPO RL agents are used.

3. An autonomous driving environment (Leurent, 2018) based on Fig. (1) for testing that our RAE in Fig. (1) is attainable in an RL setting. PPO RL agents are used.

For SFP we select the baselines to be NE (including risk/dominant payoff NE), THPE (Bielefeld, 1988) and QRE (McKelvey & Palfrey, 1995). For our iterative experiments, we select the baselines to be PSRO-{Nash, Uniform, Self-Play, THPE, QRE} where the brackets refer to the meta-solver used. In the population-based setting we believe it is fair to restrict our baselines to algorithms that operate

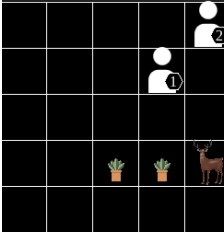

| | Num. Stag Catch | Num. Stag Gore | Num. Plant Gather |
|---|---|---|---|
| **Self-Play** | 28.81 ± 0.78 | 2.81 ± 1.00 | 1.19 ± 0.09 |
| **PSRO-Uniform** | 25.79 ± 1.24 | 6.84 ± 0.69 | 1.24 ± 0.28 |
| **PSRO-THPE** | 27.48 ± 1.22 | 4.49 ± 1.15 | 1.17 ± 0.23 |
| **PSRO-RAE (Ours)** | 0.48 ± 0.19 | 5.65 ± 0.72 | 19.87 ± 0.86 |
| **PSRO-QRE** | 26.9 ± 0.75 | 4.89 ± 0.51 | 1.24 ± 0.11 |
| **PSRO-Nash** | 28.11 ± 2.13 | 4.42 ± 0.98 | 1.09 ± 0.21 |

a) Stag-Hunt Grid-World        b) On-Equilibrium Performance

c) RAE Population vs. Nash Population

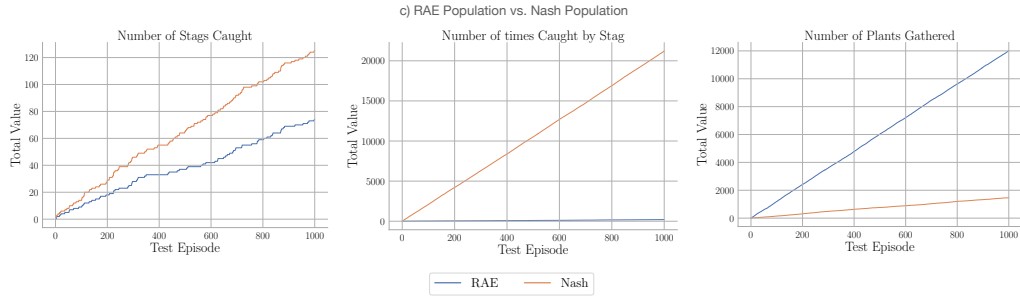

Figure 4: Stag-hunt environment results. a) Visualisation of the environment b) Results for intra-population play c) Results for RAE population vs. Nash population.

within this framework, and to not consider non-population risk-aversion algorithms and opponent modelling frameworks. In addition, our goal is to introduce and evaluate a new **game-theoretic** concept and therefore we believe the most natural comparisons are those from GT. We present our results in the form of answering three critical questions *w.r.t.* the effectiveness of RAE.

**Question 1.** Do RAE solutions have similar expected utility whilst lowering variance?

We start by investigating performance in randomly generated coordination NFGs. These NFGs are designed so that there are actions in the games that perform well if your opponent follows the same strategy but have large negative payoffs otherwise. There also exist actions that perform worse (but still positively) when coordinated on, but also maintain better performance (albeit worse than coordination) when the opponent plays a different action. We provide an example in Appendix F.

These games draw out potential pitfalls in current GT solution concepts (e.g. Nash) that focus on appealing coordination utility, without considering the variability of expected utility. We present our results in Fig. (3) where (a) represents games with 100 actions solved using SFP, and (b) represents games with 500 actions solved using our iterative framework. We plot our RAE results across multiple values of $\gamma$ in order to generate a theoretical 'efficient frontier'. An efficient frontier shows for values of expected utility what are the minimum possible variance solutions that you can find to attain said expected utility. Our figures show that, whilst our baselines are able to achieve a diverse range of expected utility values, they are unable to find the minimum variance solution which our RAE is able to find. We believe this shows the strong flexibility of our approach, in that it is able to attain any utility reward that the baselines can achieve, whilst finding a better solution in terms of variance.

**Question 2.** Can RAE act as a NE selection method?

A by-product of RAE is that it can be used as a NE selection tool. We evaluate this in a generalised stag-hunt grid world (Peysakhovich & Lerer, 2017) where there exist both 'payoff-dominant' and 'risk-dominant' NEs. We provide full environment details in Appendix (F) and provide a visualisation in Fig. (4a). There are two differing goals in the environment: 1) Collect plants on your own and receive low expected utility or 2) Hunt the stag and receive a large positive expected utility if the agents catch the stag together, and a large negative expected utility if only one agent hunts the stag. These are the 'risk-dominant' safe strategy and the 'payoff-dominant' risky strategy.

In Fig. (4) we demonstrate how RAE can effectively act as a NE selection method. In Fig. (4b) we present the expected utility where each meta-solver population is trained against itself. Notably, the final strategy of all the baselines focus on capturing the stag, which is the risky payoff-dominant strategy. However, our RAE instead finds the risk-dominant strategy in which it focuses on gathering plants and not going after the stag. The impact of this is particularly noticeable when we place an RAE population and a Nash population into the environment together as co-players, shown in Fig.

|  | Eqm Reward | Eqm Variance | Worst-Case | Num. Crashes | Num. Arrivals |
|---|---|---|---|---|---|
| **PSRO-Nash** | 0.85 ± 2.64 | 1.51 ± 0.24 | -4.84 ± 5.76 | 39.5 ± 2.12 | 10.5 ± 2.12 |
| **PSRO-Uniform** | -0.69 ± 0.87 | 1.70 ± 0.09 | -7.00 ± 2.01 | 42 ± 2.81 | 8.00 ± 2.83 |
| **PSRO-THPE** | 0.34 ± 1.29 | 1.60 ± 0.14 | -5.32 ± 3.40 | 41.5 ± 2.16 | 8.50 ± 2.12 |
| **PSRO-QRE** | 1.60 ± 0.97 | 1.44 ± 0.13 | -2.84 ± 0.94 | 43 ± 2.85 | 7.00 ± 2.85 |
| **Self-Play** | 0.97 ± 2.14 | 1.53 ± 0.12 | -4.80 ± 5.92 | 38.5 ± 4.95 | 11.50 ± 4.95 |
| **PSRO-RAE (Ours)** | 4.36 ± 2.07 | 0.33 ± 0.004 | 0.10 ± 2.68 | 5.5 ± 0.71 | 46.00 ± 1.41 |

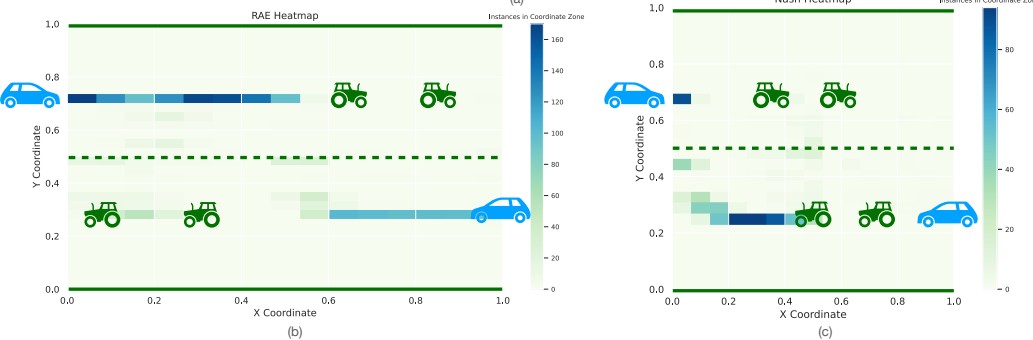

(a)

(b)

(c)

Figure 5: Results on autonomous driving environment. a) Results on 100 episodes over 5 seeds b) Position heat-map for RAE solution c) Position heat-map for Nash solution.

(4c). The Nash population still attempts to hunt the stag, but in this case the RAE population is still gathering plants therefore leading to the Nash population being caught by the stag a numerous amount of times leading to very negative expected utility. This is an overall desirable property of RAE as it suggests, in the case that RAE and NE overlap, RAE will find the risk-dominant strategy.

**Question 3. In safety-sensitive environments what sort of strategy does RAE learn to follow?**

Finally, how does RAE act in an environment where avoiding any large downside possibility is critical, for example autonomous driving. Our environment is modelled on the example in Fig. (1) where there exists two-way traffic with slow-moving vehicles and faster moving agents behind that may be interested in overtaking. From a game-theoretic standpoint this is a surprisingly difficult problem. A NE prescribes that one agent overtakes and the other waits, which is a strategy that is exposed to errors and low probability play. We provide full environment details in Appendix (F).

In Fig. (5) we provide our results. In Table (a) we provide metrics where the average value is based off of 100 episodes in the environment and the standard deviation is based over 5 different training seeds. Firstly, we note that in terms of expected utility and variance-utility RAE outperforms the baselines, whilst also maintaining strong worst-case performance. Notably, RAE arrives at a strategy that very rarely crashes, and nearly always arrives at the final destination. The same conclusion can not be drawn for any of the provided baselines.

To see why this is happening, in Fig. (4b) and (4c) we provide position heat-maps of the cars utilising the RAE strategy and the Nash strategy respectively. In the RAE heat-map one can see that the strategy taken is the safe strategy, i.e. follow behind until all vehicles in the on-coming lane have passed and then proceed to overtake. This strategy provides little expected utility for the vast majority of the episode, but remains sensitive to the risk-element of the environment which is our desired outcome. On the other hand, the Nash heat-map shows that the strategy is to overtake straight away and nearly always ends up in a crash due to car congestion in the middle of the episode.

# 8 CONCLUSION

We introduce a new risk-averse equilibrium concept, RAE, based on mean-variance analysis. Theoretically, we prove the existence and solvability of RAE and provide methods for arriving at an RAE in both small and large scale game settings. Empirically, we show that our RAE is able to locate minimum variance solutions for any expected utility, act as a NE selection method in the presence of risk-dominant equilibria and is effective at finding a safe equilibrium in a safety-sensitive autonomous driving environment. Avenues for future work should focus on the limitations of the current RAE approach, namely non-convergence guarantees in certain classes of games and the fact that RAE minimises upside and downside variance, where minimising downside variance only would be a desirable property.

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

# A  FULL PROOFS

## A.1  PROPOSITION 1 [MINIMUM VARIANCE SOLUTION]

**PROPOSITION 1.** *The solution to the optimisation (6) provides the same solutions to the following:,*

$$
\boldsymbol{\sigma}^* \in \arg\min_{\boldsymbol{\sigma}} \boldsymbol{\sigma}^T \cdot \Sigma_{\mathbf{M}} \cdot \boldsymbol{\varsigma}
$$
$$
s.t. \; \boldsymbol{\sigma}^T \cdot \mathbf{M} \cdot \boldsymbol{\sigma} \geq \mu_b
$$
$$
\sigma(a) \geq 0 \; \forall a \in A \tag{8}
$$
$$
\boldsymbol{\sigma}^T \boldsymbol{1} = 1
$$

*where $\mu_b \in \mathbb{R}$ is the lowest level of expected return that the actor is willing to accept.*

*Proof.* (Merton, 1972) shows by a Lagrange multiplier argument that the optimisation problem,

$$
\boldsymbol{\sigma}^* \in \arg\min_{\boldsymbol{\sigma}} \boldsymbol{\sigma}^T \cdot \Sigma_{\mathbf{M}} \cdot \boldsymbol{\sigma}
$$
$$
s.t. \; \boldsymbol{\sigma}^T \cdot \mathbf{M} \cdot \boldsymbol{\varsigma} \geq \mu_b
$$
$$
\sigma(a) \geq 0 \; \forall a \in A \tag{9}
$$
$$
\boldsymbol{\sigma}^T \boldsymbol{1} = 1
$$

can be rewritten as

$$
\boldsymbol{\sigma}^* \in \arg\min_{\boldsymbol{\sigma}} \boldsymbol{\sigma}^T \cdot \Sigma_{\mathbf{M}} \cdot \boldsymbol{\sigma} - \tau \Big( \boldsymbol{\sigma}^T \cdot \mathbf{M} \cdot \boldsymbol{\varsigma} \Big)
$$
$$
s.t. \; \sigma(a) \geq 0 \; \forall a \in A \tag{10}
$$
$$
\boldsymbol{\sigma}^T \boldsymbol{1} = 1
$$

which can be equivalently expressed as,

$$
\boldsymbol{\sigma}^* \in \arg\min_{\boldsymbol{\sigma}} - \Big( \boldsymbol{\sigma}^T \cdot \mathbf{M} \cdot \boldsymbol{\varsigma} - \lambda \boldsymbol{\sigma}^T \cdot \Sigma_{\mathbf{M}} \cdot \boldsymbol{\sigma} \Big)
$$
$$
s.t. \; \sigma(a) \geq 0 \; \forall a \in A \tag{11}
$$
$$
\boldsymbol{\sigma}^T \boldsymbol{1} = 1
$$

where $\lambda = \frac{1}{\tau}$.

$\square$

## A.2  THEOREM 3 [RAE EXISTENCE]

**THEOREM 3.** *For any finite N-player game where each player $i$ has a finite $k$ number of pure strategies, $A^i = \{a_1^i, ..., a_k^i\}$, an RAE exists*

*Proof.* We base our proof on Kakutani's Fixed Point Theorem

> **Lemma** (Kakutani Fixed Point Theorem). *Let $A$ be a non-empty subset of a finite dimensional Euclidean space. Let $f : A \rightrightarrows A$ be a correspondence, with $x \in A \longmapsto f(x) \subseteq A$, satisfying the following conditions:*
>
> 1. *$A$ is a compact and convex set.*
>
> 2. *$f(x)$ is non-empty for all $x \in A$.*
>
> 3. *$f(x)$ is a convex-valued correspondence: for all $x \in A$, $f(x)$ is a convex set.*

    4. $f(x)$ *has a closed graph: that is, if* $\{x^n, y^n\} \to \{x, y\}$ *with* $y^n \in f(x^n)$*, then* $y \in f(x)$.

   *Then,* $f$ *has a fixed point, that is, there exists some* $x \in A$*, such that* $x \in f(x)$.

We define our best-response function as $B_i(\boldsymbol{\sigma}_{-i}) = \arg\max_{a \in \Delta_i} r^i(a, \boldsymbol{\sigma}_{-i})$ where $u_i$ is defined as in Eq. (5) and by definition $s$ must satisfy all of the properties of a proper mixed-strategy, and the best-response correspondence is $B : \Delta \rightrightarrows \Delta$ such that for all $\boldsymbol{\sigma} \in \Delta$, we have:

$$B(\boldsymbol{\sigma}) = [B_i(\boldsymbol{\sigma}_{-i})]_{i \in N} \tag{12}$$

We show that $B(\boldsymbol{\sigma})$ satisfies the conditions of Kakutani's Fixed Point Theorem

1. $\Delta$ *is compact, convex and non-empty.*

   By definition
   $$\Delta = \Pi_{i \in N} \Delta_i \tag{13}$$
   where each $\Delta_i = \{a | \sum_j a_j = 1\}$ is a simplex of dimension $|A^i| - 1$, thus each $\Delta_i$ is closed and bounded, and thus compact. Their product set is also compact.

2. $B(\boldsymbol{\sigma})$ *is non-empty.*

   By the definition of $B_i(\boldsymbol{\sigma}_{-i})$ where $\Delta_i$ is non-empty and compact, and $r^i$ is a quadratic and hence a polynomial function in $a$. It is known that all polynomial functions are continuous, we can invoke Weirstrass's Extreme Value Theorem which states

   **Lemma.** *If a real valued-function* $f$ *is continuous on the closed interval* $[a, b]$*, then* $f$ *must attain a maximum and a minimum, each at least once. That is, there exist numbers* $c$ *and* $d$ *in* $[a, b]$ *such that:*
   $$f(c) \geq f(x) \geq f(d) \quad \forall x \in [a, b]$$

   Therefore, as $\Delta_i$ is non-empty and compact and $r^i$ is continuous in $a$, $B_i(\boldsymbol{\sigma}_{-i})$ is non-empty, and therefore $B(\boldsymbol{\sigma})$ is also non-empty.

3. $B(\boldsymbol{\sigma})$ *is a convex-valued correspondence.*

   Equivalently, $B(\boldsymbol{\sigma}) \subset \Delta$ is convex if and only if $B_i(\boldsymbol{\sigma}_{-i})$ is convex for all $i$.

   In order to show that $B_i(\boldsymbol{\sigma}_{-i})$ is convex for all $i$, we instead show that the Quadratic Programme defined by Eq. (6) is a special case of convex optimisation under certain conditions, and therefore by definition has a feasible set which is a convex set.

   A *convex optimisation problem* is one of the form,

   $$\begin{aligned} \text{minimize} \quad & f_0(x) \\ \text{s.t.} \quad & f_i(x) < 0, \ i = 1, ..., m \\ & a_i^T x = b_i, \quad i = 1, ..., p \end{aligned} \tag{14}$$

   where $f_0, ..., f_m$ are convex functions. The requirements for a problem to be a convex optimisation problem are:

   (a) the objective function must be convex
   (b) the inequality constraint functions must be convex
   (c) the equality constraint functions $h_i(x) = a_i^T x = b_i$ must be affine

   We note that a quadratic form $\mathbf{x}^T \boldsymbol{A} \mathbf{x}$ is convex if $\boldsymbol{A}$ is positive semi-definite, and strictly convex if $\boldsymbol{A}$ is positive definite (we can guarantee strict convexity by adding a small constant to the diagonal of $\boldsymbol{A}$ without impacting the variance values). In our constrained optimisation, the quadratic term $\boldsymbol{\sigma}^T \Sigma \boldsymbol{\sigma}$ is always guaranteed to be at least convex as $\Sigma$, the covariance matrix, is always at least PSD. Therefore, our objective function is convex. Additionally, it

is easy to see that our inequality constraint functions are also convex and that our equality constraint function is affine. Therefore, our Quadratic Programme is an instance of a convex optimisation problem.

Importantly, the feasible set of a convex optimisation problem is convex, since it is the intersection of the domain of the problem

$$\mathcal{D} = \bigcap_{i=0}^{m} \mathbf{dom} f_i, \tag{15}$$

, which itself is a convex set.

Therefore, for all members of the feasible set $x, y \in B_i(\boldsymbol{\sigma}_{-i})$ and all $\theta \in [0, 1]$ we have that $\theta x + (1 - \theta) y \in S$ and we have a convex-valued correspondence.

4. $B(\boldsymbol{\sigma})$ *has a closed graph.*

   Suppose to obtain a contradiction, that $B(\boldsymbol{\sigma})$ does not have a closed graph. Then, there exists a sequence $(\boldsymbol{\sigma}^n, \hat{\boldsymbol{\sigma}}^n) \to (\boldsymbol{\sigma}, \hat{\boldsymbol{\sigma}})$ with $\hat{\boldsymbol{\sigma}}^n \in B(\boldsymbol{\sigma}^n)$, but $\hat{\boldsymbol{\sigma}} \notin B(\boldsymbol{\sigma})$, i.e. there exists some $i$ such that $\hat{\boldsymbol{\sigma}}_i \notin B_i(\boldsymbol{\sigma}_{-i})$. This implies that there exists some $\boldsymbol{\sigma}'_i \in \Delta_i$ and some $\epsilon > 0$ such that

   $$r_i(\boldsymbol{\sigma}'_i, \boldsymbol{\sigma}_{-i}) > r_i(\hat{\boldsymbol{\sigma}}_i, \boldsymbol{\sigma}_{-i}) + 3\epsilon. \tag{16}$$

   By the continuity of $r_i$ and the fact that $\boldsymbol{\sigma}^n_{-i} \to \boldsymbol{\sigma}_{-i}$, we have for sufficiently large $n$,

   $$r_i(\boldsymbol{\sigma}'_i, \boldsymbol{\sigma}^n_{-i}) \geq r_i(\boldsymbol{\sigma}'_i, \boldsymbol{\sigma}_{-i}) - \epsilon. \tag{17}$$

   and combining the preceding two relations we obtain

   $$r_i(\boldsymbol{\sigma}'_i, \boldsymbol{\sigma}^n_{-i}) > r_i(\hat{\boldsymbol{\sigma}}_i, \boldsymbol{\sigma}_{-i}) + 2\epsilon \geq r_i(\hat{\boldsymbol{\sigma}}^n_i, \boldsymbol{\sigma}^n_{-i}) + \epsilon \tag{18}$$

   where the second relation follows from the continuity of $r_i$. This contradicts the assumption that $\hat{\boldsymbol{\sigma}}^n_i \in B(\boldsymbol{\sigma}^n_{-i})$ and completes the proof.

Therefore, $B(\boldsymbol{\sigma})$ satisfies the conditions of Kakutani's Fixed Point Theorem, and therefore if $\boldsymbol{\sigma}^* \in B(\boldsymbol{\sigma}^*)$ then $\boldsymbol{\sigma}^*$ is an equilibrium. $\qquad\square$

### A.3 THEOREM 4 [SFP CONVERGENCE]

**THEOREM 4.** *Given the total utility function Eq. 5 there exist RAE convergence guarantees in the category of games that are solved by SFP.*

*Proof.* We show that our utility measure can be embedded as a version of stochastic fictitious play and therefore can be used to find equilibrium in two-player zero-sum games and potential games.

A smooth fictitious play procedure is one in which the best-response, $B(\boldsymbol{\sigma})$, is derived from maximising a function of the form $r_i(\boldsymbol{\sigma}) - \lambda v_i(\boldsymbol{\sigma_i})$ where,

1. $v_i(\boldsymbol{\sigma}_i) : A_i \to \mathbb{R}$ is a strictly convex function.

2. The gradient of $v_i(\boldsymbol{\sigma}_i)$ becomes arbitrarily large near the boundary of the strategy simplex, i.e. $\lim_{\boldsymbol{\sigma}_i \to \partial A_i} |v_i(\boldsymbol{\sigma}_i)| = \infty$

which ensures that there exists a unique solution to the best-response, and that all pure strategies receive strictly positive probability in the best-response.

We have shown that our variance measure is a strictly convex objective under the assumption that $\Sigma_i$ is positive-definite. Therefore, we need to show that the gradient satisfies the boundary condition.

We start by showing that $\lim ||x_n|| = || \lim x_n ||$ if $\lim x_n = x$,

**Theorem.** *Let $X$ and $Y$ be normed spaces. If $\lim x_n = x$ in $X$ and $T : X \to Y$ is continuous, then*

$$\lim T(x_n) = T(\lim x_n)$$

*Proof.* Let $\epsilon > 0$. As $T$ is continuous, by the epsilon-delta definition of continuous functions, there exists $\delta > 0$ such that,

$$||x - y|| < \delta \Rightarrow ||T(x) - T(y)|| < \epsilon$$

As $\lim x_n = x$, there exists $n_0 \in \mathbb{N}$ such that,

$$n > n_0 \Rightarrow ||x_n - x|| < \delta$$

and it follows that,

$$n > n_0 \Rightarrow ||T(x_n) - T(x)|| < \epsilon$$

and thus

$$\lim T(x_n) = T(x) = T\lim(x_n)$$

$\square$

Since,

$$T : X \to \mathbb{R}$$
$$x \mapsto ||x||$$

is continuous, we have $\lim ||x_n|| = ||\lim x_n||$.

Next, we show that our gradient has a lower bound that satisfies the boundary condition. Note for this proof we replace $\boldsymbol{\sigma}$ with $\mathbf{x}$ and $\Sigma$ with $\text{Cov}$ as the proof relies upon the singular value decomposition and notation may become confusing.

Due to the symmetry of $\text{Cov}, \nabla_{\mathbf{x}} \text{Cov}\, \mathbf{x} = 2\,\text{Cov}\, \mathbf{x} = W$, and we show that as $\mathbf{x} \to \partial A, \lim_{\mathbf{x} \to \partial A} ||W\mathbf{x}|| > +\infty$.

$$
\begin{aligned}
\lim_{\mathbf{x} \to \partial A} ||W\mathbf{x}|| &= \lim_{\mathbf{x} \to \partial A} ||U\Sigma V^T \mathbf{x}|| \\
&= \lim_{\mathbf{x} \to \partial A} ||\Sigma V^T \mathbf{x}|| && \text{as } U \text{ is orthogonal} \\
&= \lim_{\mathbf{x} \to \partial A} ||\Sigma (V^T \mathbf{x})|| \\
&= \lim_{\mathbf{x} \to \partial A} \sum_i \sigma_i |(V^T \mathbf{x})_i| && \text{where } \sigma_i \text{ is the } i\text{-th singular value} \\
&\geq \lim_{\mathbf{x} \to \partial A} \sigma_{\min} \sum_i |(V^T \mathbf{x})_i| \\
&= \lim_{\mathbf{x} \to \partial A} \sigma_{\min} ||V^T \mathbf{x}|| \\
&= \lim_{\mathbf{x} \to \partial A} \sigma_{\min} ||\mathbf{x}|| && \text{as } V \text{ is orthogonal} \\
&= \sigma_{\min} \lim_{\mathbf{x} \to \partial A} ||\mathbf{x}|| \\
&= \sigma_{\min} || \lim_{\mathbf{x} \to \partial S} \mathbf{x}|| && \text{due to Theorem 4}
\end{aligned}
$$

At the boundary of the simplex, i.e. utilising a pure strategy, this is the specific case of a mixed-strategy where only Dirac probability distributions can be used. Therefore, in the limit there is infinite

density upon the pure strategy at the edge of the simplex and we have that $\lim_{\mathbf{x}\to\partial A}\mathbf{x} = +\infty$. We can replace this in the above,

$$\lim_{\mathbf{x}\to\partial A}||W\mathbf{x}|| \geq \sigma_{\min}||\lim_{\mathbf{x}\to\partial A}\mathbf{x}||$$
$$= \sigma_{\min}(+\infty)$$

as $\mathrm{Cov}$ is restricted to positive-definiteness, all singular values are strictly positive and we have the desired result

$$\lim_{\mathbf{x}\to\partial A}||W\mathbf{x}|| \geq +\infty \tag{19}$$

Therefore, our variance function is admissible as the perturbation function $v_i(\boldsymbol{\sigma}_i)$ in stochastic fictitious play, and retains convergence guarantees. □

## A.4  PROPOSITION 5 [SFP IS RAE]

**PROPOSITION 5.** *Suppose the SFP sequence $\{Z_t\}$ converges to $\boldsymbol{\sigma}$ in the observed strategy sense [2], then $\boldsymbol{\sigma}$ is a Risk-Averse equilibrium.*

*Proof.* Assume the observed strategy has converged to $\boldsymbol{\sigma} = (\boldsymbol{\sigma}^1, \boldsymbol{\sigma}^2)$ and that the strategy is not an RAE. This implies there exists some $\boldsymbol{\sigma}^{i,\prime}$ such that:

$$r^i(\boldsymbol{\sigma}^{i,\prime}, \boldsymbol{\sigma}^{-i}) > r^i(\boldsymbol{\sigma}^i, \boldsymbol{\sigma}^{-i}) \tag{20}$$

However, because $\boldsymbol{\sigma}$ has converged then the SFP sequence $\{Z_t\}$ will also converge such that $\lim_{t\to\infty}Z_t = \boldsymbol{\sigma}$ and because we are in an SFP process it must be the case that:

$$r^i(\boldsymbol{\sigma}^i, \boldsymbol{\sigma}^{-i}) > r^i(\boldsymbol{\sigma}^{i,\prime}, \boldsymbol{\sigma}^{-i}) \quad \forall \boldsymbol{\sigma}^{i,\prime} \in \Delta^i \tag{21}$$

and therefore $\boldsymbol{\sigma}^{i,\prime}$ can not be a best response to $\boldsymbol{\sigma}^{-i}$.

□

---

[2]Convergence in the time-average $Z_t$ does not imply convergence in the actual strategy taken at each $t$, but may for example imply cyclic actual behaviour that results in average behaviour converging.

# B SFP ROBUSTNESS

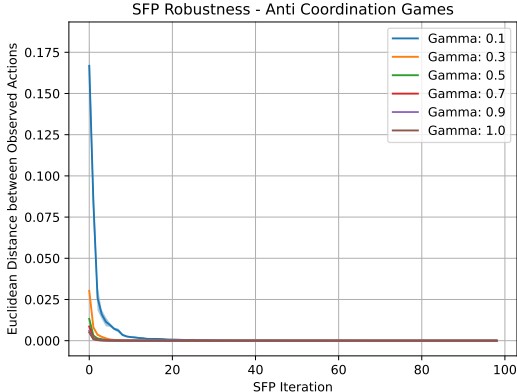

Figure 6: Euclidean distance between observed actions after each iteration on randomly generated anti-coordination games. A distance of 0 implies that the process has converged.

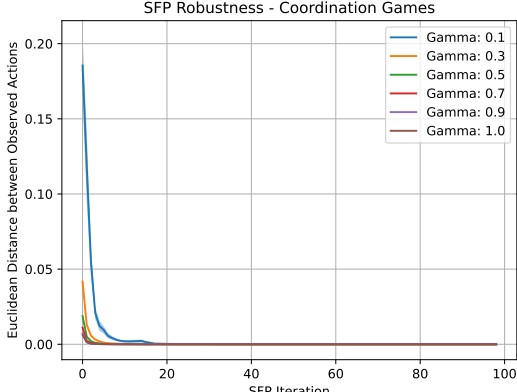

Figure 7: Euclidean distance between observed actions after each iteration on randomly generated coordination games. A distance of 0 implies that the process has converged.

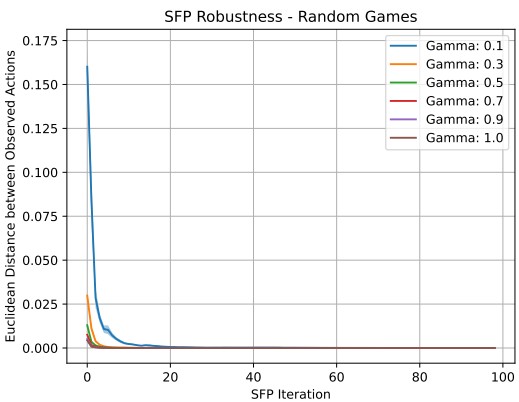

Figure 8: Euclidean distance between observed actions after each iteration on randomly generated games. A distance of 0 implies that the process has converged.

## C  Figure 3 Training Curves

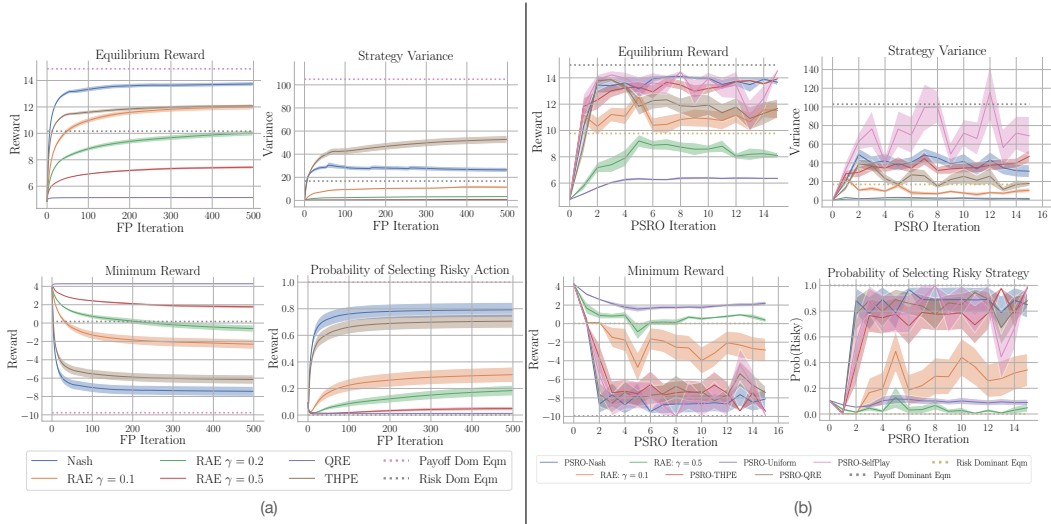

Figure 9: Training curves over multiple seeds for Figure 3.

# D   PSEUDO-CODE

---
**Algorithm 1** Iterative RAE Solver

---
1: **Initialise:** the "high-level" policy set $\Phi = \prod_{i \in \mathcal{N}} \Phi^i$
2: **for** iteration $t \in \{1, 2, ...\}$ **do**:
3:     **for** each player $i \in \mathcal{N}$ **do**:
4:         Compute meta-policy $\boldsymbol{\pi}_t$ by SFP (Eq.7).
5:         Find new policy by Oracle: $\phi_t^i = \mathcal{O}^i(\boldsymbol{\pi}_t^{-i})$.
6:         Expand $\boldsymbol{\Phi}_{t+1}^i \leftarrow \boldsymbol{\Phi}_t^i \cup \{\phi_t^i\}$.
7:         Update meta-payoff $\mathbf{M}_{t+1}$.
8: **Return:** $\boldsymbol{\pi}$ and $\boldsymbol{\Phi}$.

---

# E Hyperparameter Settings

Table 1: Hyper-parameter settings for our experiments.

| Settings | Value | Description |
|---|---|---|
| **SFP Coordination Games** | | |
| Action Dimension | 100 | Number of pure strategies available |
| FP Iterations | 100 | Number of FP belief updates |
| Tremble Probability | 0.001 | Probability of trembling to another strategy |
| Quantal Type | Softmax | Type of Quantal response equilibrium |
| # of seeds | 50 | # trials |
| **PSRO NFG Coordination Games** | | |
| Oracle method | REINFORCE | subroutine of getting oracles |
| PSRO iterations | 15 | number of PSRO iterations |
| Action Dimension | 500 | Number of pure strategies available |
| Learning rate | 0.005 | Oracle learning rate |
| Oracle Epochs | 2000 | Oracle total epochs |
| Oracle Epoch Timesteps | 100 | Timesteps per Oracle epoch |
| RAE Gamma | 0.1, 0.5 | Variance aversion parameter |
| Metasolver | RAE SFP | Metasolver method |
| Metasolver Iterations | 100 | Metasolver Iterations |
| # of seeds | 20 | # of trials |
| **Stag-Hunt Grid-World** | | |
| Oracle method | MV-PPO (Zhang et al., 2020) | subroutine of getting oracles |
| PSRO iterations | 10 | number of PSRO iterations |
| Gore Cost | 2 | Cost for getting caught by stag |
| PPO Hyperparams | Default SB3 (Raffin et al., 2021) | PPO Hyperparameter values |
| MV-PPO Variance Aversion | 0.15 | PPO Variance Aversion parameter |
| RAE Gamma | 0.15 | Variance aversion parameter |
| Metasolver | RAE SFP | Metasolver method |
| Metasolver Iterations | 100 | Metasolver Iterations |
| # of seeds | 5 | # of trials |
| **Two-Way Environment** | | |
| Oracle method | MV-PPO (Zhang et al., 2020) | subroutine of getting oracles |
| PSRO iterations | 7 | number of PSRO iterations |
| PPO Hyperparams | Default SB3 (Raffin et al., 2021) | PPO Hyperparameter values |
| MV-PPO Variance Aversion | 0.5 | PPO Variance Aversion parameter |
| RAE Gamma | 0.5 | Variance aversion parameter |
| Metasolver | RAE SFP | Metasolver method |
| Metasolver Iterations | 100 | Metasolver Iterations |
| # of seeds | 5 | # of trials |

## F ENVIRONMENTS

### F.1 RANDOMLY GENERATED NFGS

We randomly generate coordination games with $N$ actions in the following way:

---

**Algorithm 2** Iterative RAE Solver

---

1: **Initialise:** Empty $N \times N$ payoff matrix $P$
2: **for** each action $i$ **do**:
3:      Sample coordination element, $p_{ii} \sim \mathcal{U}(5, 15)$
4:      Set Payoff matrix element $P_{ii} = |p_{ii}|$
5:      **if** $P(X \leq p_{ii}) > 0.9$ **do**
6:          **for** all other actions $j$ **do**
7:              Sample anti-coordination element $p_{ij} \sim \mathcal{U}(-10, 15)$
8:              Set Payoff matrix element $P_{ij} = P_{ji} = p_{ij}$
9:      **else do**
10:          **for** all other actions $j$ **do**
11:              Sample anti-coordination element $p_{ij} \sim \mathcal{U}(0, 10)$
12:              Set Payoff matrix element $P_{ij} = P_{ji} = p_{ij}$
13: **Return:** $P$.

---

A simple 3 action example of a NFG generated following the above would be:

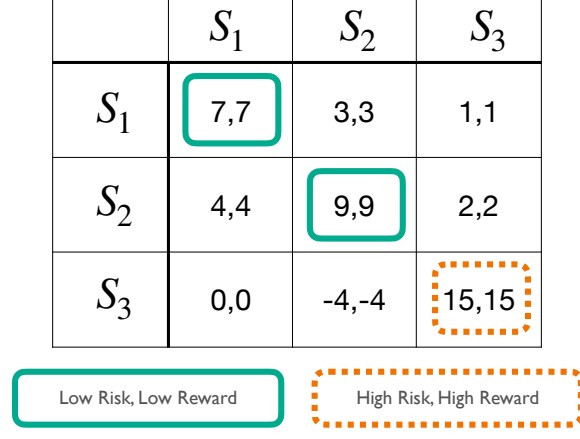

Figure 10: Game where one strategy (dotted outline) provides a high return assuming successful coordination but high variance in case the opponent does not coordinate correctly.

### F.2 STAG HUNT GRID WORLD

Our stag-hunt environment is taken from (Peysakhovich & Lerer, 2017) where we slightly alter the parameters of the game. A $5 \times 5$ grid is used with 2 players, 1 stag and 2 plants randomly spawned in. The action set of the players is $\mathcal{A} = \{\text{left}, \text{right}, \text{up}, \text{down}\}$. The stag at every time-step will move one grid space closer to the closest player on the grid, the plants do not move.

There are 3 different rewards signals in the game:

1. If a player moves over a plant they get $r = 2$ and the plant respawns elsewhere on the grid.
2. If both players move over the stag at the same time both receive $r = 5$ and the stag respawns elsewhere on the grid.
3. If a player moves over the stag on their own, or the stag moves over them on their own the player receives $r = -2$ and the stag respawns elsewhere on the grid.
4. Otherwise $r = 0$.

### F.3 AUTONOMOUS DRIVING ENVIRONMENT

Our driving environment is based on the two-way environment from (Leurent, 2018) where we make modifications to the reward function to introduce a larger factor of risk-aversion into the game. The goal of the controlled drivers is to reach the end of the road (the destination) whilst avoiding crashing and coming into too close contact with other vehicles. Slow moving drivers populate the roads moving at a constant speed of 20.

There are four reward signals in the environment:

1. If the car crashes $r = -2$.
2. If the car arrives at the destination $r = 2$.
3. If the car is travelling at a good speed ([25,30]), $r = 0.2$.
4. If the car comes very close to another car $r = -0.1$
5. Otherwise $r = 0$.

## G   COMPUTE ARCHITECTURE

All experiments run on one machine with:

- AMD Ryzen Threadripper 3960X 24 Core
- 1 x NVIDIA GeForce RTX 3090

# H    ASYMMETRIC FORMULATION

In the following section we will show the formulation of Sec. 4 but for the asymmetric case.

## H.1    UTILITY FUNCTION

Player $i$ has an action set $\mathcal{A}_i$, and a utility function $u_i$. We define the on-equilibrium utility of action $a_i^k \in \mathcal{A}_i$ against action $a_j^{k'} \in \mathcal{A}_j$ as $u(a_i^k, a_j^{k'})$ and the full utility table for player $i$ as $\mathbf{M}_i$, where the entry $\mathbf{M}_i^{k,k'}$ refers to $u(a_i^k, a_j^{k'})$ and $\mathbf{M}_i^k$ refers to $u(a_i^k, a_j^{k'}) \ \forall k'$, i.e. the vector of utilities that action $a_i^k$ receives against all other actions. Take the 2-player case, we now define the utility of the mixed-strategy for player 1 $\boldsymbol{\sigma}$ versus the mixed strategy for player 2 $\varsigma$ as

$$u(\boldsymbol{\sigma}) = \sum_{a_1^k \in A_1} \sum_{a_2^{k'} \in A_2} \sigma_k \varsigma_{k'} u(a_1^k, a_2^{k'}) = \boldsymbol{\sigma}^T \cdot \mathbf{M}_1 \cdot \varsigma. \tag{22}$$

The weighted co-variance matrix for the utility matrix $\mathbf{M}_i$ is a $|\mathcal{A}_i| \times |\mathcal{A}_j|$ matrix $\boldsymbol{\Sigma}_{\mathbf{M}_i} = [c_{kk'}]$ with entries

$$c_{kk'} = \frac{1}{1 - \sum_{i=1}^{|A_j|} \varsigma_i^2} \sum_{z=1}^{|A_j|} \varsigma_i \big( u(a_1^k, a_2^z) - \bar{\mathbf{M}}_1^k \big) \big( u(a_1^{k'}, a_2^z) - \bar{\mathbf{M}}_1^{k'} \big), \tag{23}$$

where $\bar{\mathbf{M}}_1^k = \frac{1}{|A_1|} \sum_{z=1}^{|A_2|} \sigma_k u(a_1^k, a_2^z)$, i.e. the weighted average utility for action $i$. As we are trying to minimise variance with respect to the opponent strategy we used a weighted covariance matrix such that potential variance caused by each action is weighted by its probability of selection under the opponent strategy. As will be discussed later, all actions will receive positive probability under our framework and therefore will always provide some weight in the variance calculation. This allows us to define the mixed-strategy $\boldsymbol{\sigma}$ utility variance based as follows:

$$\mathrm{Var}(\boldsymbol{\sigma}, \mathbf{M}) = \sum_{k=1}^{|A_1|} \sum_{n=1}^{|A_1|} \sigma_k \sigma_n c_{kn} = \sum_{i=1}^{|A_1|} \sigma_i^2 c_{ii} + \sum_{k=1}^{|A_1|} \sum_{n=k+1}^{|A_1|} \sigma_k \sigma_n c_{kn} = \boldsymbol{\sigma}^T \cdot \boldsymbol{\Sigma}_{\mathbf{M}_1} \cdot \boldsymbol{\sigma}. \tag{24}$$

The final utility function $r$ which considers both on- and off-equilibrium utility for strategy $\boldsymbol{\sigma}$ is,

$$r(\boldsymbol{\sigma}, \varsigma) = \boldsymbol{\sigma}^T \cdot \mathbf{M}_1 \cdot \varsigma - \gamma \big( \boldsymbol{\sigma}^T \cdot \boldsymbol{\Sigma}_{\mathbf{M}_1} \cdot \boldsymbol{\sigma} \big), \tag{25}$$

where $\gamma \in \mathbb{R}$ is the risk-aversion parameter.

## H.2    EQUILIBRIUM CONCEPT

We now define our new equilibrium concept based on the utility function (25). First start by defining the best-response map:

$$\boldsymbol{\sigma}^*(\varsigma) \in \arg\max_{\boldsymbol{\sigma}} \boldsymbol{\sigma}^T \cdot \mathbf{M}_1 \cdot \varsigma - \gamma \big( \boldsymbol{\sigma}^T \cdot \Sigma_{\mathbf{M}_1} \cdot \boldsymbol{\sigma} \big)$$
$$\text{s.t. } \sigma(a) \geq 0 \, , \forall a \in A \tag{26}$$
$$\boldsymbol{\sigma}^T \mathbf{1} = 1,$$

# I QRE FAILURE CASE

In the following section we present results on the two-action driving game described in Sec. 1 of the main article and displayed in Fig. 5.

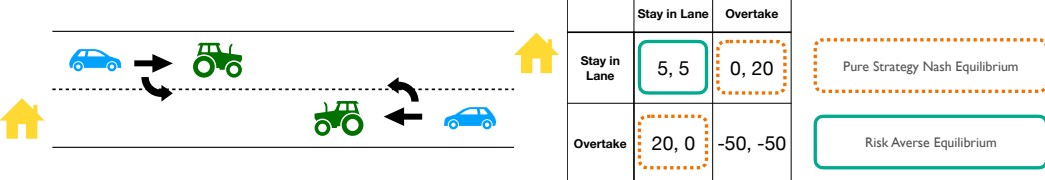

Figure 11: Two-action driving risk game.

We specifically utilise this game to show a failure case of QRE as a risk-sensitive solution. Ideally, a risk-sensitive solution concept would only play the Stay in Lane strategy as the Overtake strategy has far too high potential downside risk.

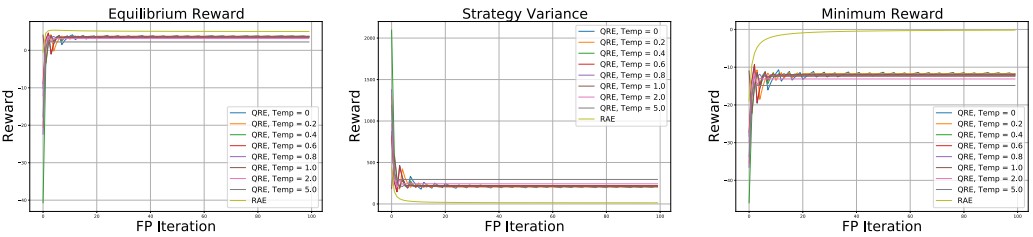

Figure 12: QRE and RAE results on two-action driving game.

As can be seen from the results in Fig. 6, for a large sample of QRE hyperparameters the equilibrium found is high variance with potential poor downside performance. We believe this is because the very large costs of the errors are easily picked up by variance analysis, but not so easily by the setup of QRE.

