# OpenReview forum: "A Risk-Averse Equilibrium for Multi-Agent Systems"
_ICLR.cc/2023/Conference — Submitted to ICLR 2023_

### Official Review · Reviewer_PRwZ · 2022-10-20

**Confidence:** 3
**Correctness:** 3
**Technical Novelty And Significance:** 3
**Empirical Novelty And Significance:** 3
**Recommendation:** 6

**Clarity, Quality, Novelty And Reproducibility:**

I found the paper overall clear and well-written, though I would suggest a proofread of the text.
As mentioned, the notion of RAE is definitely novel and the authors shed some lights on its benefits but also on its possible downsides.
Experiments are detailed, including choices of hyperparameters, which is a good sign for reproducibility upon releasing the code.

Minor:
- the joint action space is $A^1 \times ... \times A^N$ and not $A^1 =... = A^N$.
- the statement of Proposition 1 should be made clearer (e.g., $\mu_b$ appears out of nowhere).
- proof of Proposition 1 should also be spelled out more clearly as it was hard to follow. Moreover, I believe there is a typo since problem (8) and (9) have different objectives and constraints.
- it is hard to follow the sentence across pages 8 and 9, since little white space is left below caption of Fig. 5.

**Strength And Weaknesses:**

Strengths:
- The notion of RAE, to the best of my knowledge, is novel and can be interesting to the community.
- Conceptual benefits of RAEs are not only discussed but also supported by several experiments.

Weaknesses/Questions:
- The authors do not relate the proposed RAEs with its single-agent risk-averse counterparts. In which sense is this a generalization?
- Why that particular choice of augmented MDP was used to approximate rewards variance? I guess this is the main difference w.r.t. the existing methods so I believe it deserves further explanations.
- The experiments are performed with a sweep over the trade-off parameter $\gamma$. However, in practice, one does not know a reasonable value of $\gamma$ upfront and thus could reach low expected utilities as a consequence. How do you foresee $\gamma$ to be chosen? Perhaps, an adaptive approach could be used throughout training?



**Summary Of The Paper:**

The paper proposes the new notion of risk-averse equilibrium (RAE) for multi-player games, which generalizes the single agent risk-averse mean-variance framework. RAE consists of an equilibrium point of the modified game where a trade-off term representing the variance of the rewards is added to the usual expected reward. Authors prove the existence of RAEs, as well as that RAEs retain the convergent guarantees of standard stochastic fictitious play. Moreover, for large games, a population based RL approach by (Lanctot et al., 2017) is adapted to compute RAEs. Finally, several experiments are performed showing that RAEs: 1) identify minimum variance solutions for any given level of utility 2) can be used as a Nash equilibrium selection tool in the presence of a "risk-dominant" equilibrium 3) are able to find a low risk strategies in a safety-sensitive environments such as autonomous driving.

**Summary Of The Review:**

I find the paper well-motivated and the proposed notion of equilibrium relevant to the community (to the best of my knowledge, it is a novel notion). It is a very simple, yet powerful notion of risk-aversion in multi-agent scenarios, which comes with theoretical as well as practical advantages. Hence, I tend towards acceptance. However, I would be happy if the authors address the questions above. Moreover, I believe the paper would benefit from proofreading.

---

> ### Author Response · Authors · 2022-11-18
> **Response to reviewer**
>
> We thank the reviewer for taking the time to read our work and for providing some constructive comments and questions for improving the paper.
>
> **Generalisation claim -** The usage of the term generalisation in this case is probably incorrect, and we apologise to the reviewer if this is confusing. The single-agent risk-averse counterpart is the classical mean-variance portfolio selection problem, whereas our method extends this to the case of their existing another agent that impacts the utility values. In a sense, our quadratic programme could be considered a generalisation of the single-agent mean-variance optimisation by setting all $\varsigma(a_i) = 1$ so the variance matrix is no longer weighted. However, the equilibrium concept is not a natural generalisation of the mean-variance portfolio setting.
>
> **Augmented MDP -** We utilised this specific augmented MDP as we believe it is the best proxy for the total variance of a policy that we are attempting to minimise with respect to the opponent strategy. There are noted problems with minimising the exact total variance of the RL reward discussed in \cite{zhang2020meanvariance}, but it is known that minimising the variance of the per-step reward (as done by our augmented MDP) bounds the total variance of the RL reward \cite{bisi2019risk}. Therefore, our augmented MDP bounds the total variance of the RL reward in generating a new policy, which we believe is a reasonable proxy for the optimisation performed in Eq. 6.
>
> $\gamma$ **parameter -** The reviewer raises an interesting question on the selection $\gamma$. We would suggest two potential approaches, but do clarify beforehand that this is still an open question. Firstly, as $\gamma$ is agnostic to the actual game and is instead only impacted by the true utility values, one can control how these values pass into the optimisation. In particular, standardised utility values may be easier to work with as we can \textit{a priori} have a better understanding of how $\gamma$ will impact the expected return / variance on other tasks with standardised utility values.
>
> Secondly, an adaptive approach as suggested by the reviewer is a potential approach. For example, in our iterative agent generation approach, if new agents are being introduced with low expected return then the $\gamma$ value can be reduced in order to improve expected returns of new agents, whilst guaranteeing new agents are still minimum variance solutions.
>
> **Minor comments -** We thank the reviewer for pointing these out, we have updated these in the new manuscript.

---

### Official Review · Reviewer_UpJk · 2022-10-22

**Confidence:** 2
**Correctness:** 2
**Technical Novelty And Significance:** 2
**Empirical Novelty And Significance:** 3
**Recommendation:** 3

**Clarity, Quality, Novelty And Reproducibility:**

The paper is very unclear. I have asked many questions in the above section. The premise of the paper seems interesting, and I think there may be merit to this idea, but I could not figure out where that might be.

**Strength And Weaknesses:**

Broadly speaking, risk aversion is usually implemented by passing the utility function of the risk-averse player through a concave function, after which the player will become more risk-averse. That would be insufficient to handle the chicken game though-- a chicken game with utilities passed through a concave function is still a chicken game and will still have the "undesirable" equilibria. As such, I find the premise and motivating example of this paper--that there is some notion of equilibrium under which (Stay, Stay) is actually an equilibrium in a chicken game--interesting.

Unfortunately, I cannot seem to understand the technical portion of the paper, including in the motivating example, due to what appear to be errors in the writing. Here are some concrete questions, in order of appearance in the paper.


1. Eq (3-4), which seems to be absolutely critical to the paper, doesn't really make sense to me. It is also inconsistent with Eq (23-24) in Appendix H, and I'm not sure which one is the one intended by the authors.

    **a)** The denominator $1 - \sum_a \varsigma(a)$ is always zero. I'm assuming that this should be $1 - \sum_a \varsigma(a)^2$, matching (23).

    **b)** In the main body, I am assuming that the utilities $u(a_i, a_j)$ and $u(a_i, a_k)$ should instead be $u(a_j, a_i)$ and $u(a_k, a_i)$, matching (23).

    **c)** I am assuming that, in the appendix, the exclusion of the subscript $\boldsymbol\varsigma$ is a mere notational oversight and not an indication that $\mathbf{\Sigma}_{\mathbf{M}_1}$ does not actually depend on $\boldsymbol\varsigma$.

    **d)** $\mathbf{\bar M}\_i = \frac{1}{|A|} \sum_{k=1}^{|A|} \sigma_k u_i(a_i, a_k)$ (immediately after Eq (3)): assuming $\sigma_k$ should be $\varsigma_k$

    **e)** In (4) and (24), the $\sum_k \sum_{n > k}$ term should be multiplied by 2.

    With the above fixes, it seems that $\boldsymbol \sigma^\top \mathbf{\Sigma}_{\mathbf{M}, \boldsymbol\varsigma} \boldsymbol\sigma$ should be, roughly speaking, the variance that player 2 contributes to player 1's utility under profile $(\boldsymbol\sigma, \boldsymbol\varsigma)$, scaled by $1 / (1 - \lVert \boldsymbol\varsigma \rVert_2^2)$. Also, it is unclear to me what the intuition is behind the scaling factor. I would appreciate the authors explicitly including extra discussion surrounding the equilibrium notion.

1. Even the fixed formulation would divide by zero when $\lVert \boldsymbol\varsigma \rVert_2^2 = 1$, i.e., when $\boldsymbol\varsigma$ is a pure strategy. In particular, that seems to happen in the example calculation at the end of Sec 4.1. The authors do write "all actions will still receive positive probability in our framework", but the example is clearly is not fully mixed. What is the calculation used in the example to conclude that (Stay, Stay) is an equilibrium? In any case, for the rest of my review I will assume that $\boldsymbol\varsigma$ is actually fully mixed and therefore there is no division by zero.

1. The use of the variance in Eq (5), in particular instead of standard deviation, seems rather strange to me, as it implies that the resulting notion of equilibrium will not be scale-invariant: a player whose utilities are halved will suddenly find herself essentially less risk-averse, because the variance will be cut by 1/4. This feels like it requires further motivation.

1. In Proposition 1, what is $u_b$? Eq (6) contains no $u_b$ nor an obvious place to insert one.

1. Theorem 4's proof sketch in the paper body seems wrong: $\text{Var}(\boldsymbol\sigma, \boldsymbol\varsigma, \mathbf M)$ is quadratic in $\boldsymbol\sigma$, and therefore cannot possibly have arbitrarily large gradient at the boundary of a compact set (namely, a simplex). Also, I have no a priori reason to believe that the variance is *strictly* convex (as requried by SFP), merely convex. Am I missing something?


Relatively minor things not affecting my score:

1. Related work, second paragraph, "suggesting strategic uncertainty is by" - seems like an unfinished sentence.
1. $\sigma_k$ and $\sigma(a_k)$ are both used to mean the probability that a mixed strategy $\sigma$ assigns to action $k$. In the interest of cleanliness, I'd advise picking one of the two and sticking with it.
1. Indices $i, j, k$ are used to index both players and actions at various points in the paper. This is unnecessarily confusing; I'd advise again using indices to mean only one thing. For example, $i, j$ can index players, and $a, a'$ can index actions.
1. The theorem numbering in the appendix is inconsistent with the numbering in the main body. This is unnecessarily confusing. For example, Theorems 3 and 4 in the main body are proven as Theorems 4 and 5 in the appendix, respectively.

**Summary Of The Paper:**

The paper defines a notion of risk-averse equilibrium, develops some theory surrounding the notion, and runs some experiments illustrating that their notion achieves a better trade-off between utility and variance than other notions.

**Summary Of The Review:**

An interesting premise, but lack of clarity and seemingly numerous critical typos made the paper impossible to parse for me. I would be willing to read a revised version that more clearly illustrates the main idea.

---

> ### Author Response · Authors · 2022-11-18
> **Response to reviewer**
>
> We thank the reviewer for the constructive comments and critiques of the work, and hope we can alleviate some of their concerns here.
>
> **1 b-e** - All of the suggested changes from the reviewer are correct, and we apologise for missing these details as they understandably can make our work more confusing. We have made all these changes in the revised version.
>
> **1 a Scaling Factor** - We apologise to the reviewer for the confusion here. This is actually a mistake, and there should not be any scaling factor in the calculation. All of the influence of the strategy of the other player is included in $\varsigma(a_i)$. We have updated this in a new version of the manuscript.
>
> **2 -** We have discussed the scaling factor above, but will comment on the second half of this question. In the example, our statement that the strategy profiles we are comparing are totally pure is incorrect which we apologise for. As we state earlier in the paper, we do treat all strategy profiles as fully mixed (emulating the idea that errors may happens) and such it is better to define the strategy profiles as: $\boldsymbol{S}_1 = ((1-\epsilon, 0+\epsilon), (1-\epsilon, 0+\epsilon))$ and $\boldsymbol{S}_2 = ((0+\epsilon, 1-\epsilon), (1-\epsilon, 0+\epsilon))$. For the case of the example, we used $\epsilon = 0.01$. The goal of the example was to show that, whilst $\boldsymbol{S}_2$ is not strictly a Nash equilibrium, it is arbitrarily close to one and even when the probability of taking the dangerous action is arbitrarily small it still has a large enough impact on variance to warrant the (Stay, Stay) equilibrium (which is also fully mixed).
>
> **3 Scale Invariance -** We first note for the reviewer that neither standard deviation nor variance are scale-invariant in terms of the utilities, but it is fair to look at how the values scale in respect to constants (SD scales exactly with the constant, whereas Var scales by the square of the constant). We agree that the squared scaling factor could potentially be problematic, however we believe the pros of utilising variance outweigh this con. The major reason that we would suggest for using variance is that it more heavily weights outliers from the mean in comparison to the standard deviation. The intended usage of this equilibrium is to control for the potential of large negative falls in utility from low probability strategies, and therefore we believe utilising a measure that heavily weights outliers is more apt for our cause. We have added this motivation to the manuscript
>
> **4 $\mu_b$ -** We refer the reviewer to Appendix A to fully understand proposition 1. $\mu_b$ is meant to represent \textit{any} expected return value, and is utilised in a different optimisation in Appendix A (not in Eq. 6) that has an identical solution set. The purpose of proposition 1 is to state that the optimisation of Eq. 6 provides the minimum variance solution for any expected return value ($\mu_b$), and the level of expected return is controlled by $\gamma$.
>
> **5 Theorem 4 -** We believe it is important for the reviewer to look at Appendix A to understand the proof as the proof sketch is very brief.
>
> In terms of the strict convexity of the variance, the reviewer is correct that the variance matrix is positive semi-definite and therefore only convex. It is easy to transform the variance matrix from positive semi-definite to positive definite (to induce strict convexity) by adding a small constant to the diagonal elements (which has no impact on the actual variance values). We have made this clear in the new manuscript.
>
> **Minor comments -** We thank the reviewer for the minor comments which we have cleaned up in the new manuscript.
>
> We hope with the above that we have been able to address all of the concerns of the reviewer and that the paper looks much better. If the reviewer agrees, could they please consider updating their score?

---

> > ### Comment · Reviewer_UpJk · 2022-11-19
> > **Response**
> >
> > Thanks for the reply.
> >
> > 1. Thank you. I think there is another error: $\bar{\mathbf{M}}_i$ should not have a $1/|A|$ in its definition. Somehow I didn't catch this in my previous review.
> >
> > 2. Unless I'm missing something, the variance calculation is now a continuous function of the mixed strategy vectors $\boldsymbol\sigma$ and $\boldsymbol\varsigma$. As such, I do not see the purpose of taking $\epsilon$-trembling strategies and taking a limit $\epsilon \to 0$---this should achieve the same result as directly setting $\epsilon = 0$. If the only way to achieve the desired result is to take strictly positive $\epsilon$, I do not think the paper then achieves its stated goal in this example---after all, in my view, the whole point of the paper with respect to the chicken example is to recover some notion in which (Stay, Stay)---as a pure strategy profile---is an equilibrium. If we need to insist on some mixing anyway, we may as well instead work with some more classical techniques for risk aversion, e.g., (as I mentioned), passing rewards through a concave function.
> >
> > 3. To be clear, by "scale-invariant" I meant that if you multiply the utilities for a single player by a constant, then the equilibria should not change--that is a property not held by the risk-averse notion in the present paper, and the fact that the property does not hold perplexes me, especially when a simple modification removes the issue.
> >
> > 4. In the interest of keeping the main body of the paper self-contained, I strongly recommend the authors to place any necessary information about $\mu_b$ in the main body.
> >
> > 5. Appendix A only confirms my fears. There seems to be an error: at the top of page 17, $\mathbf x$ is a mixed strategy---in particular, it is a vector of $\ell_1$ norm exactly 1, and cannot have infinite limit in any direction. Indeed, as I said in my original review, it is not possible for a quadratic function to achieve infinite gradient on a compact set.
> >
> > Due to the above issues, especially (2), (3), and (5), I think the paper is still insufficient for acceptance in its current state, and therefore I keep my score.

---

### Official Review · Reviewer_1b8K · 2022-10-23

**Confidence:** 3
**Correctness:** 4
**Technical Novelty And Significance:** 2
**Empirical Novelty And Significance:** 2
**Recommendation:** 5

**Clarity, Quality, Novelty And Reproducibility:**

**Clarity:**
- Overall, the paper is well-written and easy to follow.
- However, the authors did not seem to mention the approach of extending the two-player game case into the general multi-player games, and I think more emphasis is needed as the paper title says "multi-agent systems".

**Novelty:**
- Many proofs in the paper, for example, the convergence results, seem to depend heavily on the results and conclusions in the existing literature, and it is not sure to me what is the theoretical novelty in the proofs of these results.

**Strength And Weaknesses:**

**Strength:**
- The authors considered an interesting and common risk-averse learning scenario where safety-aware strategies are desired. To the best of my knowledge, the case where the utility function measures the potential variance of the opponents' strategy has not been well-studied in the existing literature. Traditional methods, such as quantal response equilibrium, are not able to resolve such tasks perfectly.
- A learning method based on the stochastic fictitious play is proposed for the equilibrium, and a theoretical guarantee is provided as long as the SFP sequence converges.
- An RL-based learning algorithm is available to handle large games, where the empirical performance of the algorithm is promising. It offers a way to find min variance strategies for a given expected utility and select Nash equilibrium in the risk-dominant equilibria.

**Weakness:**
- The authors showed the existence of the mean-variance equilibrium, but the uniqueness of the risk-averse equilibrium is still lacking. Which equilibrium those proposed algorithms converge to is not clear to me.
- Proposition 1 shows that there exists $\gamma$ such that the solution is the min variance solution. My question is: how do we choose the parameter $\gamma$ if we would like to prevent some extreme behavior to happen? This does not seem to be obvious for large games.
- There are no theoretical guarantees for equilibrium learning with iterative agent generation.

**Summary Of The Paper:**

The paper introduces a risk-averse mean-variance equilibrium to multi-agent games, and the authors also showed the existence of such a risk-averse equilibrium. A fictitious play type of learning algorithm is proposed to solve for the equilibrium and a corresponding RL-based approximation algorithm enjoys good empirical performance on a set of real-world learning scenarios.

**Summary Of The Review:**

The paper studies an important and relevant topic of multi-agent game learning with risk-averse equilibrium, and the authors proposed their algorithms for learning such equilibrium on real-world data. However, some of the theoretical guarantees for both the equilibrium and the algorithm are still lacking.

===========================

I tend to agree with the reviewers WyXC and UpJk that the paper requires more improvement at the current stage. Therefore, I lower my score to 5.

---

> ### Author Response · Authors · 2022-11-18
> **Response to reviewer**
>
> We thank the reviewer for taking the time to read our work and for providing some constructive comments and questions for improving the paper.
>
> **Uniqueness** -  We agree with the reviewer that the uniqueness of the equilibrium is a missing aspect of our work. Whilst each optimisation of the mean-variance programme is unique (assuming positive definiteness of the variance matrix), we do not know if the initial conditions of the system may lead to different final equilibrium solutions. This is an open question.
>
> $\gamma$ **in Proposition 1** -  We consider $\gamma$ to be a hyperparameter that needs to be tuned for the desired performance of the practitioner. The selection of $\gamma$ is specifically impacted by the magnitude of the utilities, so standardisation of the utilities may make it easier to select a value of $\gamma$. However, we have not looked specifically into this in this work.

---

### Official Review · Reviewer_WyXC · 2022-11-01

**Confidence:** 4
**Correctness:** 1
**Technical Novelty And Significance:** 2
**Empirical Novelty And Significance:** 2
**Recommendation:** 3

**Clarity, Quality, Novelty And Reproducibility:**

The lack of discussion and comparison to other research in the area of risk averse game theory puts this paper clearly below the bar.

**Strength And Weaknesses:**

Strengths:
1) The paper examines a very interesting setting that combines many agents (decision-making) as well as risk-averse agents (i.e. non-standard expected utility maximizers)

Weaknesses:
1) The authors are not aware of a lengthly line of research that examines answers to exactly these type of questions on the intersection of economics and computation. I do not believe that the paper should be considered for publication before its finding are put in their correct context.

**Summary Of The Paper:**

The paper claims to be amongst the first paper's to consider the interplay of games and risk aversion in decision making and examines existence of a solution/equilibrium concept as well as methods for computing of approximating them.

"Despite the importance of risk-aversion in the single-agent decision making literature (Zhang et al., 2020; Mihatsch & Neuneier, 2002; Chow et al., 2017), it is surprising that this research idea is not lively in the current game theory research domain."

This is a patently false statement. Below, I put forward a number of recent papers in top game theory conferences and journals which are directly working on the intersection of game theory and risk averse agents. None of these papers are cited.

Thanasis Lianeas, Evdokia Nikolova, Nicolas E. Stier Moses.
"Risk-averse selfish routing"
Mathematics of Operations Research, vol. 44(1), pp. 38-57, 2019.

Georgios Piliouras, Evdokia Nikolova and Jeff S. Shamma.
"Risk Sensitivity of Price of Anarchy under Uncertainty."
ACM Transactions on Economics and Computation (TEAC), Volume 5, Issue 1, November 2016, Article No. 5.

Evdokia Nikolova, Nicolas E. Stier Moses.
"The Burden of Risk Aversion in Mean-Risk Selfish Routing"
In Proceedings of the Sixteenth ACM Conference on Economics and Computation (EC'15). Portland, OR, June 15-19, 2015.

Hota, Ashish R., Siddharth Garg, and Shreyas Sundaram.
"Fragility of the commons under prospect-theoretic risk attitudes."
Games and Economic Behavior 98 (2016): 135-164.

Angelidakis, Haris, Dimitris Fotakis, and Thanasis Lianeas.
"Stochastic congestion games with risk-averse players."
International Symposium on Algorithmic Game Theory. Springer, Berlin, Heidelberg, 2013.

Fiat, Amos, and Christos Papadimitriou.
"When the players are not expectation maximizers."
International Symposium on Algorithmic Game Theory. Springer, Berlin, Heidelberg, 2010.

This is a far from exhaustive list with several of these papers having dozens of citations. I believe the authors need to carefully examine the related literature and put their work in the right context.

**Summary Of The Review:**

Interesting subject matter but the authors are clearly unaware of a lot of relevant work. The paper is not ready for publication without comparing/contrasting with these prior approaches.

---

> ### Author Response · Authors · 2022-11-18
> **Response to reviewer**
>
> We thank the reviewer for taking the time to read our work and for providing a list of papers from the risk-averse game-theory literature. We agree that the exact phrase 'it is surprising that this research idea is not lively in the current game theory research domain' is incorrect (which has now been removed from the manuscript), as shown by the reviewer, and have added the following to our related work section:
>
> In terms of risk-aversion outside of equilibrium concepts, competitive network games \cite{wardrop1952road} and the non expected utility maximising setting \cite{fiat2010players} have been studied the most. Risk aversion in the network setting is based on a generalisation of the classical selfish routing model \cite{beckmann1956studies} to incorporate uncertain delays. \cite{nikolova2014mean} consider a mean-variance framework for Wardrop equilibria in this setting, whilst \cite{lianeas2019risk} extend this research to looking at how risk aversion degrades the performance of a routing system. Whilst the mean-variance approach is the same underlying notion as our work, we instead propose a solution for general games rather than routing games. In general games, \cite{fiat2010players} remove the assumption of expectation maximisation and show that under risk averse utility functions there may exist no Nash equilibria. Further works have generally focused on Price of Anarchy \cite{piliouras2016risk,kesselheim2018price}, which study how removing the assumptions of risk neutrality in general games impacts the difference between the achieved worst equilibrium and the maximum possible welfare of the system. Our work follows a similar strand in looking at general games, but focuses on defining a new equilibrium concept rather than establishing how risk-averse agents change the convergence properties to classical equilibrium concepts. In addition, we frame our work such that it is more scalable for usage alongside RL techniques.
>
> Based on the above addition to the paper, which frames our work more extensively in the wider literature, we hope that the reviewer would consider revising their score.

---

> > ### Comment · Reviewer_WyXC · 2022-11-19
> > **Response**
> >
> > I thank the authors for taking into my comments and adding the related references to their paper, but I still think there is no quick fix to this and that the introduction of the paper has to be rewritten. The current introduction still says nothing about all of this considerable amount of work, which is itself not exhaustive. Hence, I think the framing and justification of the current work is not complete.  For example, I believe that the mean variance approach is also followed in \cite{piliouras2016risk} under the name second moment method and they show that as a result the price of anarchy deteriorates from a small constant to unbounded. This is clearly evidence that this specific treatment of risk/uncertainty has rather undesirable properties and thus the paper examines a variety of different approaches that circumvent these problems. In the presence of these results that need to be discussed in more detail in the introduction, one now has to give more detailed justification about why this specific approach to risk is pursued versus e.g. one of the other six alternatives explored in that paper. And the above line of reasoning follows just by taking a single of these papers into more deep consideration.
> >
> > Overall, I find the direction pursued by the authors rather interesting but I believe that understanding how their work fits into this larger framework still requires some work that should significantly affect how the whole paper is written instead of just a passing reference.

---

### Author Response · Authors · 2022-11-18
**Response to all reviewers**

We thank all of the reviewers for taking the time to provide constructive criticism of the our paper! We have provided an updated manuscript where we have made any requested changes from the reviewers. The biggest changes are highlighted in blue.

---

### Decision · Program_Chairs · 2023-01-20

**Decision:**

Reject

**Justification For Why Not Higher Score:**

It just ignores prior work.

**Justification For Why Not Lower Score:**

N/A

**Metareview: Summary, Strengths And Weaknesses:**

This paper fails to cite a veritable litany of prior work in this field. It must be completely reworked to place itself properly in the context of the extant body of work.